# Learning Once, Routing Right: Information-Theoretic Gating for Online Continual Mixture-of-Experts

## Abstract

Continual Learning (CL) requires models to acquire knowledge from streaming data without sacrificing past performance. Mixture-of-Experts (MoE) architectures address this challenge through dynamic expert allocation, yet their gating mechanisms remain heuristically designed and theoretically underexplored. To bridge this gap, we analyze the role of gating strategies in shaping the Minimum Excess Risk (MER) under the online continual learning setting. Our key theoretical contribution reveals a novel connection: the MER is closely related to the mutual information between expert assignments and labels/outputs. Building on this theoretical foundation, we design two novel loss functions grounded in mutual information, applicable to both fully labeled and label-free scenarios. To further guarantee computational efficiency, we develop a lightweight, matrix-based estimator with a rigorous joint entropy formulation. Through extensive evaluations on MNIST, Fashion-MNIST, KMNIST, and EMNIST, our approach consistently outperforms SOTA baselines, reducing overall error by up to 12.3% and forgetting by 3.9%, both with statistical significance.

## 1 Introduction

Continual Learning (CL) addresses the challenge of learning from sequential tasks without revisiting past data. The key obstacle is catastrophic forgetting—the erosion of prior knowledge when adapting to new information (Wang et al., 2024b). Existing research largely falls into three categories: regularization-based methods (Benzing, 2022; Lin et al., 2022) that constrain parameter updates, architecture-based approaches (Kang et al., 2022; Douillard et al., 2022) that dedicate separate capacity to different tasks, and replay strategies (Tiwari et al., 2022; Gopalakrishnan et al., 2022) that explicitly store and reuse past experiences. While these methods have advanced the field, they rely on task-specific assumptions or incur prohibitive computational costs, highlighting the need for more principled and scalable alternatives.

Among potential alternatives, the Mixture-of-Experts (MoE) framework has emerged as a particularly promising candidate. By routing inputs through a sparse set of experts, MoE enables efficiency and specialization (Chi et al., 2022; Zadouri et al., 2023), and it has already proven essential in scaling large language models (Lin et al., 2024; Xue et al., 2024). Recent work has begun to explore its use in continual learning, including expert training (Rypeść et al., 2024), capacity expansion (Yu et al., 2024), and theoretical perspectives (Li et al., 2025; Li & Duan, 2025). Unlike traditional CL methods that rely on regularization, parameter isolation, or replay, MoE is uniquely governed by its gating mechanism, which dynamically assigns data to experts and directs the trajectory of knowledge across tasks. However, existing approaches (Li et al., 2024; 2025; Li & Duan, 2025) treat gating heuristically, prioritizing load balancing or convergence rather than developing a principled foundation for its continual optimization.

To address this gap, we investigate the optimization of gating behavior in MoE under the Online Continual Learning (OCL) setting, where each sample is observed only once (Guo et al., 2022; Wang et al., 2024a). Neuroscientific studies (Fang et al., 2025) highlight the role of specific brain structures in gating conscious perception, underscoring the importance of context-dependent routing. This insight closely parallels MoE gating under streaming conditions. Although Minimum Excess Risk

(MER) (Xu & Raginsky, 2022; Hafez-Kolahi et al., 2023) quantifies the discrepancy between oracle and learned expert performance, its value is critically determined by how effectively the gating mechanism routes data to experts under limited information.

Guided by this perspective, we develop a rigorous theoretical framework to analyze MoE models in the OCL setting. First, we derive an upper bound on the MER and reveal its tight connection to the mutual information between expert assignments and labels/outputs. This finding provides a principled account of how gating governs continual learning, motivating a reformulation of MER minimization as the maximization of mutual information. Building on this foundation, we introduce two mutual information–based loss functions tailored to both fully labeled and label-free scenarios. To support efficient and accurate optimization, we further propose a lightweight matrix-based estimator equipped with a rigorous joint entropy formulation.

To validate our approach, we conduct extensive experiments on MNIST, Fashion-MNIST, KMNIST, and EMNIST. Our method consistently and significantly surpasses existing SOTA baselines, achieving up to 12.3% lower MoE error and 3.9% reduced forgetting, both supported by rigorous statistical significance (one-tailed Z-tests with $Z = -29.6$ and $Z = -15.1$, respectively). Our contributions can be summarized as follows:

- We establish a rigorous framework for analyzing MoE under OCL, showing that Minimum Excess Risk (MER) is fundamentally tied to the mutual information between expert assignments and labels/outputs.
- Building on this framework, we design two novel loss functions tailored to fully labeled and label-free regimes, supported by a lightweight, matrix-based mutual information estimator with a principled entropy formulation.
- Through comprehensive evaluations, we demonstrate that our method is both efficient and effective, delivering substantial gains over all baselines across multiple benchmarks.

## 2 RELATED WORKS

Mitigating catastrophic forgetting has been a central challenge in Continual Learning (CL) (Wang et al., 2024b). Existing approaches largely fall into three categories. Regularization-based methods constrain the update of parameters deemed important for past tasks, thereby trading off stability and plasticity (Kirkpatrick et al., 2017; Benzing, 2022; Wang et al., 2021; Lin et al., 2022). Architecture-based methods isolate parameters across tasks through dynamic expansion or modularization, effectively reducing interference (Xue et al., 2022; Kang et al., 2022; Hyundong & Kim, 2022; Douillard et al., 2022). Replay-based methods maintain a memory buffer of samples or gradients and rehearse them during training, directly counteracting forgetting (Chaudhry et al., 2019; Tiwari et al., 2022; Gong et al., 2022; Gopalakrishnan et al., 2022).

In parallel, the Mixture-of-Experts (MoE) paradigm has emerged as a scalable architecture for deep learning (Wang & Van Hoof, 2022; Zhou et al., 2022; Chi et al., 2022; Zadouri et al., 2023). By activating only a sparse subset of experts per input, MoE achieves both specialization and efficiency, which has fueled its adoption in large-scale systems, notably large language models (Du et al., 2022; Li et al., 2024; Lin et al., 2024; Xue et al., 2024). The ability of MoE to handle diverse and heterogeneous data distributions makes it a natural candidate for continual and multi-task learning scenarios.

The intersection of MoE and CL has recently gained attention (Wang et al., 2022; Doan & Seyed Iman Mirzadeh, 2023; Rypeść et al., 2024; Yu et al., 2024). For instance, SEED (Rypeść et al., 2024) partitions data based on distributional dissimilarity, training experts on minimally overlapping subsets to enhance robustness. DDAS (Yu et al., 2024) expands the capacity of vision-language models through MoE, mitigating forgetting in cross-modal continual settings. On the theoretical front, Li et al. (2025) analyze expert specialization and gating control in CL, uncovering new convergence behaviors, while Li & Duan (2025) extend MoE theory to mobile edge computing with adaptive gating to improve continual performance under dynamic environments.

A distinguishing challenge in this line of work is the gating mechanism, which dynamically routes data to experts and shapes the overall learning trajectory. Unlike classical CL concerns (e.g., regularization, replay, parameter isolation), gating introduces a new optimization dimension. Yet most

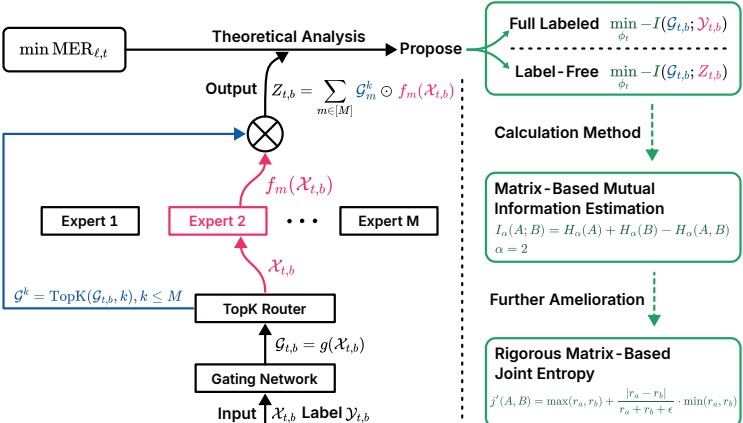

Figure 1: An illustration of the MoE model in our methods.

existing approaches (Fedus et al., 2022; Li et al., 2024; 2025; Li & Duan, 2025) treat it heuristically, emphasizing load balancing or convergence stability. In contrast, we pursue a principled, information-theoretic formulation of MoE gating under the OCL setting.

## 3 METHODS

### 3.1 PROBLEM FORMULATION

We consider the Online Continual Learning (OCL) setting with $T$ training rounds. Let $\mathcal{D} = (X_t, Y_t)_{t \in [T]}$ denote the sequential dataset, where each task is associated with a distinct distribution $p(X_t, Y_t)$ and disjoint label set $Y_i \cap Y_j = \emptyset$ for $i \neq j$, consistent with the class-incremental learning setup. Let $b \in \mathcal{B}_t$ be the batch index, and $\mathcal{B}_t$ denote the set of batch indices in the $t$-th training round. In each round $t$, the stream provides mini-batches $(X_{t,b}, Y_{t,b})_{b \in \mathcal{B}t}$, which are processed once in a single-pass manner. We maintain a replay buffer using reservoir sampling of size $n$, yielding augmented batches $(\mathcal{X}_{t,b}, \mathcal{Y}_{t,b})$ for training.

Given an incoming $(\mathcal{X}_{t,b}, \mathcal{Y}_{t,b})$, we formulate the inference process of MoE models as shown in Figure 1. For $M$ experts, let $\mathcal{G}_{t,b} \in \mathbb{R}^{n \times M}$ be the gating output of $g : \mathcal{X}_{t,b} \to \mathcal{G}_{t,b}$. In the top-$k$ sparse gating setting, we define the intermediate $\mathcal{G}^k$ and the final MoE output $Z_{t,b}$ as:

$$\mathcal{G}_{t,b} = g(\mathcal{X}_{t,b}); \mathcal{G}^k = \text{TopK}(\mathcal{G}_{t,b}, k), k \leq M; Z_{t,b} = \sum_{m \in [M]} \mathcal{G}_m^k \odot f_m(\mathcal{X}_{t,b}), \mathcal{G}_m^k \in \mathbb{R}^{n \times 1}, \quad (1)$$

where $f_m : \mathcal{X}_{t,b} \to Z_m, Z_m \in \mathbb{R}^{n \times C}$ denotes the $m$-th expert, $C$ denotes the number of classes. $\text{TopK}(\mathcal{G}_{t,b}, k)$ retains the top-$k$ values per row of $\mathcal{G}_{t,b}$ and sets the remaining entries to zero.

Let $\ell_e(\cdot; \theta_t)$ be the task loss of the expert networks with parameters $\theta_t$ and $\ell_g(\cdot; \phi_t)$ be the gating regularization loss with gating parameters $\phi_t$, we jointly optimize:

$$\min_{\theta_t, \phi_t} \ell_e(Z_{t,b}, \mathcal{Y}_{t,b}; \theta_t) + \ell_g(\mathcal{G}_{t,b}, Z_{t,b}, \mathcal{Y}_{t,b}; \phi_t). \quad (2)$$

In this paper, we focus on the design of $\ell_g(\cdot; \phi_t)$.

For MoE in the OCL setting, we define the error-based metrics. Let $\ell_{\theta_t, \phi_t}(Z_\tau, \mathcal{Y}_\tau), \tau \in [C]$ be the error on the test set of $\tau$-th class, where $\theta_t$ represents the weights after the $t$-th training rounds. The overall error is given by:

$$\mathcal{E}_t = \frac{1}{C} \sum_{\tau \in [C]} \ell_{\theta_t, \phi_t}(Z_\tau, \mathcal{Y}_\tau), \quad (3)$$

and the overall forgetting rate is defined as:

$$\mathcal{F}_t = \frac{1}{C} \sum_{\tau \in [C]} \left( \ell_{\theta_t, \phi_t}(Z_\tau, \mathcal{Y}_\tau) - \min_{t' < t} \left\{ \ell_{\theta_{t'}, \phi_{t'}}(Z_\tau, \mathcal{Y}_\tau) \right\} \right). \quad (4)$$

Lower values of $\mathcal{E}_t$ and $\mathcal{F}_t$ indicate better overall performance and knowledge retention, respectively.

### 3.2 MINIMIZE MINIMUM EXCESS RISK BY MUTUAL INFORMATION MAXIMIZATION

The Minimum Excess Risk (MER) is the gap between the minimum expected loss attainable by learning from data and the minimum expected loss that could be achieved if the model realization were known. MER captures the inherent uncertainty and limitations of a model's performance by quantifying the gap between the best achievable outcome through data-driven learning and the ideal performance attainable with perfect knowledge of the true model parameters. Thus, by optimizing the gating behavior to improve data allocation, the MER of the MoE model can be minimized.

The definition of MER in Bayesian learning is established in existing work (Xu & Raginsky, 2022). The Bayes risk with respect to a loss function $\ell$ is defined as:

$$R_\ell(\mathcal{Y}_t|\mathcal{X}_t, \mathcal{D}_{1:t}) = \inf_{f:\mathcal{X}_t \to Z_t|\mathcal{D}_{1:t}} \mathbb{E}\left[\ell\left(f(\mathcal{X}_t), \mathcal{Y}_t\right)\right],$$

and the fundamental limit of the Bayes risk is given by:

$$R_\ell(\mathcal{Y}_t|\mathcal{X}_t, \theta_t) = \inf_{f:\mathcal{X}_t \to Z_t|\theta_t} \mathbb{E}\left[\ell\left(f(\mathcal{X}_t), \mathcal{Y}_t\right)\right].$$

The MER definition is:

$$\mathrm{MER}_{\ell,t} = R_\ell(\mathcal{Y}_t|\mathcal{X}_t, \mathcal{D}_{1:t}) - R_\ell(\mathcal{Y}_t|\mathcal{X}_t, \theta_t).$$

It quantifies the regret incurred by the best data-driven decision rule, which lacks access to the true model parameters, relative to the optimal omniscient decision rule, thereby capturing the intrinsic difficulty of the learning problem. We extend the MER definition to MoE models in OCL as:

$$R_\ell(\mathcal{Y}_{t,b}|\mathcal{X}_{t,b}, \mathcal{G}_{t,b}, \mathcal{D}_{1:t}) = \inf_{f:\mathcal{X}_{t,b} \to Z_{t,b}|\mathcal{G}_{t,b}, \mathcal{D}_{1:t}} \mathbb{E}\left[\ell\left(f(\mathcal{X}_{t,b}), \mathcal{Y}_{t,b}\right)\right];$$

$$R_\ell(\mathcal{Y}_{t,b}|\mathcal{X}_{t,b}, \mathcal{G}_{t,b}, \theta_t) = \inf_{f:\mathcal{X}_{t,b} \to Z_{t,b}|\mathcal{G}_{t,b}, \theta_t} \mathbb{E}\left[\ell\left(f(\mathcal{X}_{t,b}), \mathcal{Y}_{t,b}\right)\right];$$

$$\mathrm{MER}_{\ell,t} = R_\ell(\mathcal{Y}_{t,b}|\mathcal{X}_{t,b}, \mathcal{G}_{t,b}, \mathcal{D}_{1:t}) - R_\ell(\mathcal{Y}_{t,b}|\mathcal{X}_{t,b}, \mathcal{G}_{t,b}, \theta_t).$$

$\mathcal{G}_{t,b}$ depends on $\mathcal{X}_{t,b}$, included here to emphasize the role of gating in MER evaluation. Then, we derive an upper bound for MER in terms of $I(\mathcal{G}_{t,b}; \mathcal{Y}_{t,b})$:

**Proposition 1.** *For the zero-one loss,*

$$\mathrm{MER}_{\ell^{01},t} \le \sqrt{\frac{1}{2}\left(I(\theta_t; \mathcal{Y}_{t,b}) + I(\mathcal{X}_{t,b}; \mathcal{Y}_{t,b}) - I(\mathcal{G}_{t,b}; \mathcal{Y}_{t,b})\right)}.$$

Please refer to our Appendix A for the proof of Proposition 1. Similarly, this proposition can be generalized to arbitrary loss functions $\ell$ (Xu & Raginsky, 2022):

$$\mathrm{MER}_{\ell,t} \le \varphi^{-1}\left(I(\theta_t; \mathcal{Y}_{t,b}) + I(\mathcal{X}_{t,b}; \mathcal{Y}_{t,b}) - I(\mathcal{G}_{t,b}; \mathcal{Y}_{t,b})\right),$$

where $\varphi^{-1}$ is monotonic increase and depends on the choice of $\ell$.

Note that $I(\mathcal{X}_{t,b}; \mathcal{Y}_{t,b})$ depends solely on the data distribution and remains constant during optimization. The expert loss function minimize $I(\theta_t; \mathcal{Y}_{t,b})$, which remains independent of the gating network optimization process. Thus, we only focus on the term $-I(\mathcal{G}_{t,b}; \mathcal{Y}_{t,b})$. Minimizing the MER is equivalent to:

$$\min_{\phi_t} -I(\mathcal{G}_{t,b}; \mathcal{Y}_{t,b}).$$

Focusing on the inference process, we have the Markov data processing chain $\mathcal{G}_{t,b} \leftarrow \mathcal{X}_{t,b} \to \mathcal{Y}_{t,b}$. Considering the backpropagation of expert networks with $\ell(Z_{t,b}, \mathcal{Y}_{t,b})$, we have $\mathcal{Y}_{t,b} \to \theta_t \leftarrow Z_{t,b}$ and re-arrange it as: $\mathcal{G}_{t,b} \to \mathcal{X}_{t,b} \to \mathcal{Y}_{t,b} \to \theta_t \to Z_{t,b}$. According to the well-known data processing inequality (DPI), we obtain: $-I(\mathcal{G}_{t,b}; \mathcal{Y}_{t,b}) \le -I(\mathcal{G}_{t,b}; Z_{t,b})$. Then, we provide the MER optimization without label data:

$$\min_{\phi_t} -I(\mathcal{G}_{t,b}; Z_{t,b}).$$

Finally, we propose two gating loss functions to minimize MER through mutual information maximization during training after accessing the buffer pool:

$$\mathcal{L}_Y(\mathcal{G}_{t,b}, Z_{t,b}, \mathcal{Y}_{t,b}; \theta_t, \phi_t) = \ell_e(Z_{t,b}, \mathcal{Y}_{t,b}; \theta_t) + \gamma_y \cdot \frac{H(\mathcal{Y}_{t,b}) - I(\mathcal{G}_{t,b}; \mathcal{Y}_{t,b})}{H(\mathcal{Y}_{t,b})}, \gamma_y \in \mathbb{R}^+;$$

$$\mathcal{L}_Z(\mathcal{G}_{t,b}, Z_{t,b}, \mathcal{Y}_{t,b}; \theta_t, \phi_t) = \ell_e(Z_{t,b}, \mathcal{Y}_{t,b}; \theta_t) + \gamma_z \cdot \frac{H(Z_{t,b}) - I(\mathcal{G}_{t,b}; Z_{t,b})}{H(Z_{t,b})}, \gamma_z \in \mathbb{R}^+,$$

(5)

where the cross entropy loss $\ell_e$ can be any expert loss function. The denominators $H(\mathcal{Y}_{t,b})$ and $H(Z_{t,b})$ normalize the values to $[0, 1]$, promoting numerical stability across datasets.

### 3.3 LIGHTWEIGHT MATRIX-BASED MUTUAL INFORMATION ESTIMATION

Computing mutual information via traditional eigenvalue-based methods requires the full eigenspectrum of a Gram matrix, which grows with the number of samples and incurs $O(n^3)$ time complexity (Dong et al., 2023), making it infeasible for high-dimensional or large-scale data. Thus, we propose a lightweight method with a reduced time complexity of $O(n^2)$.

For a given continuous random variable $x \in \mathbb{R}^d$ that values in a finite set $\mathbf{X}$, let $\kappa : \mathbf{X} \times \mathbf{X} \to \mathbb{R}$ be a real-valued positive kernel that is also infinitely divisible (Dong et al., 2023). Given $\{x_i\}_1^n \subset \mathbf{X}$ and the Gram matrix $K$ obtained from $K_{ij} = \kappa(x_i, x_j)$, a matrix-based analogue to Rényi's $\alpha$-entropy can be defined as (Giraldo et al., 2014):

$$H_\alpha(A) = \frac{1}{1 - \alpha} \log_2 (\text{tr}(A^\alpha)),$$

where $A_{ij} = \frac{1}{n} \frac{K_{ij}}{\sqrt{K_{ii} K_{jj}}}$ is a normalized kernel matrix. To accommodate varying values in the OCL setting, we calculate the mean $\mu_X$ and standard deviation $\sigma_X$ of $X = \{x_i\}_1^n$. Then, we propose a calculation pipeline for the 2-order matrix-based Rényi entropy:

$$\hat{X} = \{\hat{x}_i\}_1^n = \frac{X - \mu_X}{\sigma_X \cdot d^{\frac{1}{4}}}; A_{ij} = \frac{\kappa(\hat{x}_i, \hat{x}_j)}{n}, \kappa(\hat{x}_i, \hat{x}_j) = \exp\left(-\frac{\|\hat{x}_i - \hat{x}_j\|_2^2}{2\sigma^2}\right);$$

$$H_\alpha(A) = -\log_2\left(\text{tr}(A^2)\right) = -\log_2 \sum_{i=1}^n \sum_{j=1}^n A_{ij}^2 = -\log_2 \sum_{i=1}^n \sum_{j=1}^n \frac{1}{n^2} \exp\left(-\frac{\|\hat{x}_i - \hat{x}_j\|_2^2}{\sigma^2}\right)$$

(6)

$$= -\log_2\left(\frac{1}{n} + \frac{2}{n^2} \sum_{i=1}^n \sum_{j>i}^n \exp\left(-\frac{\|\hat{x}_i - \hat{x}_j\|_2^2}{\sigma^2}\right)\right), \alpha = 2, \sigma = 1,$$

where $\hat{X} = \frac{X - \mu_X}{\sigma_X \cdot d^{\frac{1}{4}}}$ ensures the data behavior aligns with $\mu = 0, \sigma = 1, d = 1$ in the calculation of $H_\alpha(A)$ for numerical stability in high dimensions. Given that $I_\alpha(A; B) = H_\alpha(A) + H_\alpha(B) - H_\alpha(A, B)$, we first define the traditional 2-order joint entropy:

$$\hat{X}_A = \{\hat{x}_i^a\}_1^n = \frac{X_A - \mu_{X_A}}{\sigma_{X_A} \cdot d^{\frac{1}{4}}}; \hat{X}_B = \{\hat{x}_i^b\}_1^n = \frac{X_B - \mu_{X_B}}{\sigma_{X_B} \cdot d^{\frac{1}{4}}}; j(A, B) = \|\hat{x}_i^a - \hat{x}_j^a\|_2^2 + \|\hat{x}_i^b - \hat{x}_j^b\|_2^2;$$

$$H_\alpha(A, B) = H_\alpha\left(\frac{A \circ B}{\text{tr}(A \circ B)}\right) = -\log_2 \sum_{i=1}^n \sum_{j=1}^n \frac{1}{n^2} \exp\left(-\frac{1}{\sigma^2} j(A, B)\right).$$

However, when $A = B$, we observe that $\|\hat{x}_i^a - \hat{x}_j^a\|_2^2 = \|\hat{x}_i^b - \hat{x}_j^b\|_2^2 \geq 0$, implying $H_\alpha(A, A) \geq H_\alpha(A)$. To address this issue, we propose a new 2-order joint entropy:

$$r_a = \|\hat{x}_i^a - \hat{x}_j^a\|_2^2, r_b = \|\hat{x}_i^b - \hat{x}_j^b\|_2^2, \epsilon > 0; j'(A, B) = \max(r_a, r_b) + \frac{|r_a - r_b|}{r_a + r_b + \epsilon} \cdot \min(r_a, r_b);$$

$$H_\alpha'(A, B) = -\log_2 \sum_{i=1}^n \sum_{j=1}^n \frac{1}{n^2} \exp\left(-\frac{1}{\sigma^2} j'(A, B)\right),$$

(7)

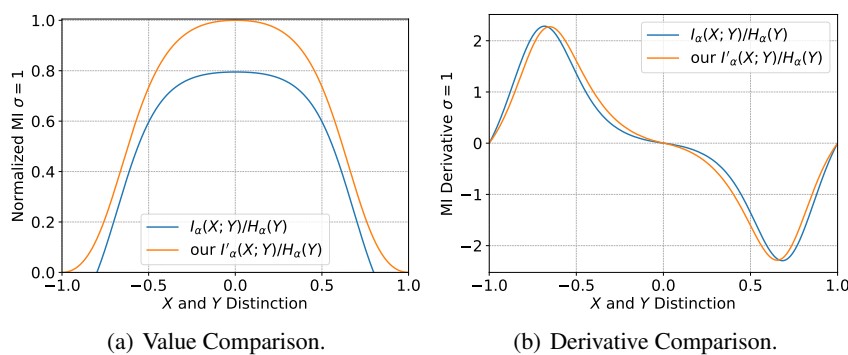

(a) Value Comparison.

(b) Derivative Comparison.

Figure 2: Mutual information estimation comparison.

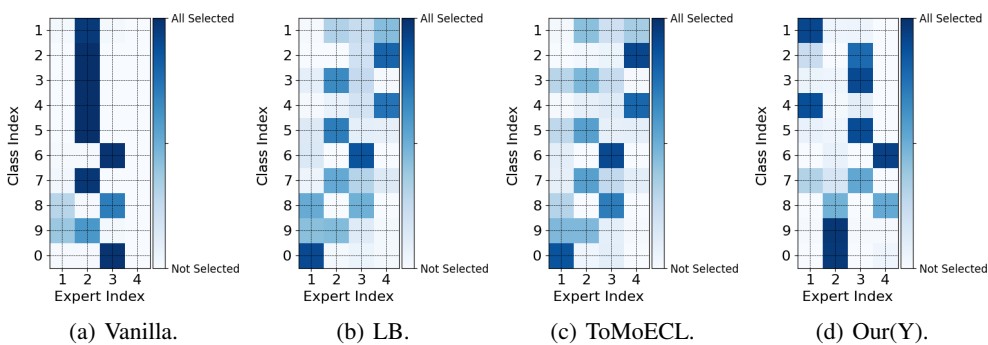

(a) Vanilla.

(b) LB.

(c) ToMoECL.

(d) Our(Y).

Figure 3: Example gating behavior comparison on MNIST ($M = 4, k = 1$).

where $\max(r_a, r_b) \leq j'(A, B) \leq j(A, B)$. This construction guarantees that when $A = B$, the joint entropy reduces exactly to $H_\alpha(A)$, mitigating the overestimation of self-information while preserving pairwise distinctions for $A \neq B$. Finally, we define the new matrix-based mutual information estimation as:

$$I'_\alpha(A; B) = H_\alpha(A) + H_\alpha(B) - H'_\alpha(A, B), \alpha = 2, \tag{8}$$

which provides a numerically stable and computationally efficient estimate suitable for high-dimensional data.

Figure 2 compares the proposed estimator with the traditional matrix-based MI. By fixing $Y$ and varying $X$, we observe that the proposed estimator maintains a well-defined value range and exhibits larger gradients near extrema, facilitating stable and efficient optimization.

### 3.4 COMPUTATIONAL COMPLEXITY ANALYSIS

The main computational cost of our method comes from estimating the matrix-based mutual information, $I(\mathcal{G}_{t,b}; \mathcal{Y}_{t,b})$ or $I(\mathcal{G}_{t,b}; Z_{t,b})$, where $\mathcal{G}_{t,b} \in \mathbb{R}^{n \times M}$ and $Z_{t,b} \in \mathbb{R}^{n \times C}$. To compute meaningful distances for class labels, we convert $\mathcal{Y}_{t,b}$ into its one-hot encoding $\mathcal{Y}'_{t,b} \in \mathbb{R}^{n \times C}$. Typically, the number of experts satisfies $M \leq C$, as using more experts than classes is generally unnecessary (Li et al., 2025).

Under these settings, the time complexity of our method is $O(n^2(C + M)) = O(n^2C)$. In our experiments, we use $n = 1000$ and $C = 10$, which reduces the training time from days (as required by traditional $O(n^3C)$ eigenvalue-based methods) to a few minutes. Compared to the baseline, the additional overhead is only about 30 seconds. Unlike classical approaches relying on eigen-decomposition or matrix multiplication, our method consists solely of element-wise operations with broadcasting, making it highly compatible with modern GPU acceleration and yielding significant speedups in practice.

| Methods | $k \in \{1,1,1\}$ for $M \in \{1,4,7\}$ | | | $k \in \{1,2,3\}$ for $M \in \{1,4,7\}$ | | | $k \in \{1,4,7\}$ for $M \in \{1,4,7\}$ | | |
|---|---|---|---|---|---|---|---|---|---|
| Vanilla | 25.0±2.0 | 24.7±2.1 | 22.5±1.3 | 25.0±2.0 | 26.8±1.6 | 23.1±1.6 | 25.0±2.0 | 22.0±2.4 | 20.8±1.7 |
| LB | 25.0±2.0 | 30.8±1.2 | 28.7±0.91 | 25.0±2.0 | 28.6±1.6 | 27.0±1.3 | 25.0±2.0 | 30.7±1.3 | 27.8±1.3 |
| ToMoECL | 25.0±2.0 | 31.0±1.3 | 29.0±1.0 | 25.0±2.0 | 30.5±1.6 | 27.0±1.2 | 25.0±2.0 | 31.0±1.3 | 28.0±1.3 |
| ToMoEMEC | 25.0±2.0 | 25.7±1.7 | 21.8±0.7 | 25.0±2.0 | 26.9±1.9 | 20.3±1.0 | 25.0±2.0 | 22.5±2.4 | 19.8±1.7 |
| Our(Y) + LB | 25.0±2.0 | 29.9±1.3 | 30.3±0.88 | 25.0±2.0 | 28.9±1.6 | 26.5±1.2 | 25.0±2.0 | 28.3±1.2 | 27.8±1.3 |
| Our(Z) + LB | 25.0±2.0 | 30.3±0.91 | 29.2±0.71 | 25.0±2.0 | 29.7±1.6 | 26.9±0.95 | 25.0±2.0 | 28.0±1.5 | 28.9±0.56 |
| Our(Y) w/o JE | 25.0±2.0 | 20.2±0.83 | 20.4±0.89 | 25.0±2.0 | 22.5±2.1 | 16.4±1.8 | 25.0±2.0 | 21.8±1.4 | 20.9±1.7 |
| Our(Z) w/o JE | 25.0±2.0 | 17.3±0.69 | 17.0±0.91 | 25.0±2.0 | 21.4±1.6 | 19.5±2.3 | 25.0±2.0 | 25.8±1.6 | 19.5±1.6 |
| Our(Y) | 25.0±2.0 | 16.7±1.3 | 20.2±1.3 | 25.0±2.0 | 17.3±1.2 | 15.7±1.3 | 25.0±2.0 | **21.2±1.4** | 18.9±1.3 |
| Our(Z) | 25.0±2.0 | **15.8±0.67** | **16.7±0.96** | 25.0±2.0 | 21.1±1.9 | 18.6±1.6 | 25.0±2.0 | 22.8±0.82 | **18.7±1.4** |

Table 1: MoE overall error (%) on MNIST, 10 runs.

| Methods | $k \in \{1,1,1\}$ for $M \in \{1,4,7\}$ | | | $k \in \{1,2,3\}$ for $M \in \{1,4,7\}$ | | | $k \in \{1,4,7\}$ for $M \in \{1,4,7\}$ | | |
|---|---|---|---|---|---|---|---|---|---|
| Vanilla | 52.4±1.8 | 38.2±2.2 | 36.0±3.1 | 52.4±1.8 | 38.5±4.1 | 36.6±1.9 | 52.4±1.8 | 45.9±2.5 | 37.0±1.4 |
| LB | 52.4±1.8 | 54.5±0.67 | 55.3±0.36 | 52.4±1.8 | 52.6±0.74 | 52.6±0.45 | 52.4±1.8 | 49.9±1.4 | 49.3±1.6 |
| ToMoECL | 52.4±1.8 | 54.4±0.48 | 55.1±0.51 | 52.4±1.8 | 53.2±0.61 | 53.2±0.50 | 52.4±1.8 | 49.5±1.5 | 49.5±1.5 |
| ToMoEMEC | 52.4±1.8 | 38.9±2.2 | 36.3±2.0 | 52.4±1.8 | 37.5±3.1 | 35.1±2.4 | 52.4±1.8 | 44.9±1.2 | 36.6±1.9 |
| Our(Y) + LB | 52.4±1.8 | 54.0±0.67 | 56.2±0.49 | 52.4±1.8 | 52.8±0.79 | 53.6±0.55 | 52.4±1.8 | 47.0±1.8 | 49.0±2.0 |
| Our(Z) + LB | 52.4±1.8 | 54.2±0.53 | 55.7±0.43 | 52.4±1.8 | 52.9±0.85 | 53.0±0.46 | 52.4±1.8 | 47.5±1.8 | 45.7±1.7 |
| Our(Y) w/o JE | 52.4±1.8 | 34.2±2.1 | 34.7±1.9 | 52.4±1.8 | 36.1±1.8 | 37.1±1.6 | 52.4±1.8 | 43.6±3.1 | 35.6±1.7 |
| Our(Z) w/o JE | 52.4±1.8 | 39.7±2.5 | 35.8±2.6 | 52.4±1.8 | 40.3±2.7 | 35.9±2.2 | 52.4±1.8 | 40.9±2.0 | 36.2±2.2 |
| Our(Y) | 52.4±1.8 | 33.8±2.1 | **29.6±1.2** | 52.4±1.8 | **34.8±2.6** | 34.1±2.0 | 52.4±1.8 | 41.9±2.4 | 35.3±1.7 |
| Our(Z) | 52.4±1.8 | 33.8±1.9 | 32.1±1.1 | 52.4±1.8 | 38.3±3.2 | **31.9±3.2** | 52.4±1.8 | **39.5±1.5** | 34.6±2.5 |

Table 2: MoE overall error (%) on Fashion-MNIST, 10 runs.

## 4 EXPERIMENTS

### 4.1 EXPERIMENT SETUP

We conduct extensive experiments to validate the effectiveness of our proposed method. Four benchmark datasets are used: MNIST (LeCun et al., 2010), Fashion-MNIST (Xiao et al., 2017), KM-NIST (Clanuwat et al., 2018), and EMNIST (Cohen et al., 2017). To simulate a streaming setting, data arrive in small batches corresponding to a subset of classes, with a reservoir buffer of size 1000 employed for replay. Specifically, for MNIST, Fashion-MNIST, and KMNIST, the 10 classes are divided into 5 groups, with each batch containing 16 samples from each of two classes (32 samples total). For EMNIST, the 47 balanced classes are split into 8 groups, with each batch providing 16 samples from up to 6 classes. Each class undergoes 400 data arrivals.

We vary the number of experts $M \in 1, 4, 7$ and evaluate three top-$k$ sparsity configurations: (1) fixed $k = 1$, (2) $k$ increasing with $M$ as $k = 1, 2, 3$, and (3) $k = M$ (no sparsity). Each setup is repeated 10 times, and statistical significance is assessed.

Our method is compared against several baselines: Load-Balance (LB) (Fedus et al., 2022; Shazeer et al., 2017; Li et al., 2024), ToMoECL (Li et al., 2025), and ToMoEMEC (Li & Duan, 2025). We also conduct further studies to assess component contributions: integrating a load-balancing strategy ("Our + LB"), removing the joint entropy function ("Our w/o JE"), and using a standard loss $\ell_{ce}$ without our proposed modifications ("Vanilla").

The gating network consists of a single-layer fully connected network, while each expert network has three fully connected layers with 512 hidden units, followed by a softmax output. The gating network outputs are used for routing, and the expert networks are trained with the cross-entropy loss. Optimization is performed with RAdam (Liu et al., 2020). Learning rates and mutual information coefficients are set as follows: for MNIST and Fashion-MNIST, 0.0002 and $\gamma_y = \gamma_z = 2$; for KMNIST and EMNIST, 0.0001 and $\gamma_y = \gamma_z = 0.1$.

Performance is measured in terms of overall error and forgetting rate, following the definitions in Section 3.1. Specifically, $\ell_{\theta_t, \phi_t}$ is implemented as the 0-1 loss. All experiments are conducted on an Ubuntu server equipped with eight NVIDIA RTX 4090 GPUs.

| Methods | $k \in \{1,1,1\}$ for $M \in \{1,4,7\}$ | | | $k \in \{1,2,3\}$ for $M \in \{1,4,7\}$ | | | $k \in \{1,4,7\}$ for $M \in \{1,4,7\}$ | | |
|---|---|---|---|---|---|---|---|---|---|
| Vanilla | $52.6_{\pm0.99}$ | $40.5_{\pm1.1}$ | $38.5_{\pm1.1}$ | $52.6_{\pm0.99}$ | $40.4_{\pm1.3}$ | $37.5_{\pm1.2}$ | $52.6_{\pm0.99}$ | $39.8_{\pm1.1}$ | $38.2_{\pm1.0}$ |
| LB | $52.6_{\pm0.99}$ | $53.2_{\pm0.69}$ | $54.3_{\pm0.56}$ | $52.6_{\pm0.99}$ | $51.4_{\pm0.63}$ | $51.9_{\pm0.70}$ | $52.6_{\pm0.99}$ | $51.2_{\pm0.73}$ | $51.5_{\pm0.93}$ |
| ToMoECL | $52.6_{\pm0.99}$ | $53.2_{\pm0.69}$ | $54.3_{\pm0.56}$ | $52.6_{\pm0.99}$ | $51.5_{\pm0.69}$ | $52.0_{\pm0.74}$ | $52.6_{\pm0.99}$ | $51.3_{\pm0.73}$ | $51.6_{\pm0.91}$ |
| ToMoEMEC | $52.6_{\pm0.99}$ | $40.3_{\pm1.2}$ | $39.6_{\pm1.3}$ | $52.6_{\pm0.99}$ | $40.5_{\pm1.3}$ | $36.7_{\pm1.3}$ | $52.6_{\pm0.99}$ | $40.0_{\pm0.71}$ | $38.1_{\pm0.73}$ |
| Our(Y) + LB | $52.6_{\pm0.99}$ | $52.7_{\pm0.63}$ | $54.3_{\pm0.36}$ | $52.6_{\pm0.99}$ | $51.4_{\pm0.75}$ | $51.8_{\pm0.67}$ | $52.6_{\pm0.99}$ | $51.7_{\pm0.84}$ | $51.3_{\pm0.86}$ |
| Our(Z) + LB | $52.6_{\pm0.99}$ | $53.5_{\pm0.68}$ | $54.7_{\pm0.41}$ | $52.6_{\pm0.99}$ | $51.8_{\pm0.81}$ | $51.8_{\pm0.63}$ | $52.6_{\pm0.99}$ | $50.2_{\pm1.1}$ | $51.1_{\pm0.81}$ |
| Our(Y) w/o JE | $52.6_{\pm0.99}$ | $39.3_{\pm1.0}$ | $38.0_{\pm1.1}$ | $52.6_{\pm0.99}$ | $38.6_{\pm0.93}$ | $36.5_{\pm0.82}$ | $52.6_{\pm0.99}$ | $\underline{39.4}_{\pm1.3}$ | $37.7_{\pm1.1}$ |
| Our(Z) w/o JE | $52.6_{\pm0.99}$ | $40.0_{\pm1.1}$ | $38.2_{\pm1.4}$ | $52.6_{\pm0.99}$ | $37.6_{\pm0.86}$ | $38.0_{\pm0.88}$ | $52.6_{\pm0.99}$ | $39.8_{\pm0.85}$ | $38.7_{\pm1.0}$ |
| Our(Y) | $52.6_{\pm0.99}$ | $\mathbf{38.1}_{\pm\mathbf{0.67}}$ | $37.7_{\pm0.88}$ | $52.6_{\pm0.99}$ | $\mathbf{37.1}_{\pm\mathbf{0.69}}$ | $36.3_{\pm0.82}$ | $52.6_{\pm0.99}$ | $\mathbf{39.0}_{\pm\mathbf{1.0}}$ | $36.4_{\pm1.1}$ |
| Our(Z) | $52.6_{\pm0.99}$ | $\underline{38.7}_{\pm0.77}$ | $\mathbf{37.0}_{\pm\mathbf{0.76}}$ | $52.6_{\pm0.99}$ | $\underline{37.5}_{\pm0.72}$ | $\mathbf{36.3}_{\pm\mathbf{0.73}}$ | $52.6_{\pm0.99}$ | $39.7_{\pm1.2}$ | $\mathbf{35.9}_{\pm\mathbf{0.51}}$ |

Table 3: MoE overall error (%) on KMNIST, 10 runs.

| Methods | $k \in \{1,1,1\}$ for $M \in \{1,4,7\}$ | | | $k \in \{1,2,3\}$ for $M \in \{1,4,7\}$ | | | $k \in \{1,4,7\}$ for $M \in \{1,4,7\}$ | | |
|---|---|---|---|---|---|---|---|---|---|
| Vanilla | $72_{\pm0.67}$ | $67.2_{\pm1.3}$ | $67.7_{\pm1.2}$ | $72_{\pm0.67}$ | $66.5_{\pm0.54}$ | $64.8_{\pm1.8}$ | $72_{\pm0.67}$ | $68.5_{\pm1.1}$ | $66.5_{\pm0.82}$ |
| LB | $72_{\pm0.67}$ | $74.6_{\pm0.72}$ | $74.2_{\pm0.48}$ | $72_{\pm0.67}$ | $74.6_{\pm0.64}$ | $75.5_{\pm0.74}$ | $72_{\pm0.67}$ | $74.1_{\pm0.73}$ | $72.1_{\pm0.71}$ |
| ToMoECL | $72_{\pm0.67}$ | $74.8_{\pm0.73}$ | $74.5_{\pm0.48}$ | $72_{\pm0.67}$ | $74.5_{\pm0.84}$ | $75.5_{\pm0.73}$ | $72_{\pm0.67}$ | $74.1_{\pm0.73}$ | $72.1_{\pm0.71}$ |
| ToMoEMEC | $72_{\pm0.67}$ | $66.5_{\pm1.2}$ | $67.8_{\pm1.2}$ | $72_{\pm0.67}$ | $66.3_{\pm0.71}$ | $62.9_{\pm1.0}$ | $72_{\pm0.67}$ | $68.1_{\pm1.1}$ | $66.0_{\pm0.75}$ |
| Our(Y) + LB | $72_{\pm0.67}$ | $74.9_{\pm0.83}$ | $74.3_{\pm0.48}$ | $72_{\pm0.67}$ | $74.7_{\pm0.83}$ | $75.7_{\pm0.68}$ | $72_{\pm0.67}$ | $73.9_{\pm0.76}$ | $71.8_{\pm0.76}$ |
| Our(Z) + LB | $72_{\pm0.67}$ | $75.2_{\pm0.84}$ | $74.5_{\pm0.47}$ | $72_{\pm0.67}$ | $74.4_{\pm0.67}$ | $75.4_{\pm0.60}$ | $72_{\pm0.67}$ | $73.9_{\pm0.75}$ | $72.0_{\pm0.78}$ |
| Our(Y) w/o JE | $72_{\pm0.67}$ | $60.2_{\pm1.0}$ | $55.7_{\pm0.60}$ | $72_{\pm0.67}$ | $\underline{61.3}_{\pm0.95}$ | $\underline{58.1}_{\pm0.50}$ | $72_{\pm0.67}$ | $66.6_{\pm1.4}$ | $64.9_{\pm1.0}$ |
| Our(Z) w/o JE | $72_{\pm0.67}$ | $66.0_{\pm0.73}$ | $64.8_{\pm1.0}$ | $72_{\pm0.67}$ | $65.5_{\pm1.6}$ | $63.7_{\pm1.2}$ | $72_{\pm0.67}$ | $66.6_{\pm1.0}$ | $64.1_{\pm1.0}$ |
| Our(Y) | $72_{\pm0.67}$ | $\mathbf{59.4}_{\pm\mathbf{0.63}}$ | $\mathbf{55.4}_{\pm\mathbf{0.54}}$ | $72_{\pm0.67}$ | $\mathbf{60.3}_{\pm\mathbf{1.2}}$ | $\mathbf{55.7}_{\pm\mathbf{0.98}}$ | $72_{\pm0.67}$ | $\mathbf{66.1}_{\pm\mathbf{1.4}}$ | $64.1_{\pm1.1}$ |
| Our(Z) | $72_{\pm0.67}$ | $64.9_{\pm1.1}$ | $63.6_{\pm0.79}$ | $72_{\pm0.67}$ | $65.3_{\pm1.2}$ | $62.8_{\pm1.2}$ | $72_{\pm0.67}$ | $\underline{66.2}_{\pm1.2}$ | $\mathbf{64.0}_{\pm\mathbf{1.3}}$ |

Table 4: MoE overall error (%) on EMNIST (Balanced), 10 runs.

## 4.2 EVALUATION RESULTS

**MNIST**  Figure 3 compares gating behaviors. The vanilla method fails to distribute tasks across experts, while other baselines fail to concentrate classes on single experts. Our method achieves both balanced allocation and class-level specialization. Table 1 reports that "Our(Y)" achieves up to a $9.5\%$ reduction in MoE error over all baselines ($Z = -15.02$, $p < 2.7\mathrm{e}{-}51$), while "Our(Z)" attains $8.9\%$ improvement ($Z = -12.77$, $p < 1.3\mathrm{e}{-}37$). Performance deteriorates when combined with LB ($Z = -9.62$, $p < 3.4\mathrm{e}{-}22$) or when the joint entropy is ablated ($Z = -4.93$, $p < 4.1\mathrm{e}{-}7$), confirming the necessity of adaptive gating and entropy optimization. Table 5 further shows up to a $3.9\%$ reduction in forgetting rate.

**Fashion-MNIST**  As shown in Table 2, "Our(Y)" reduces MoE error by $6.4\%$ ($Z = -6.09$, $p < 5.7\mathrm{e}{-}10$), while "Our(Z)" achieves the same $6.4\%$ reduction ($Z = -6.94$, $p < 2.0\mathrm{e}{-}12$). Integration with LB causes significant degradation ($Z = -5.38$, $p < 3.8\mathrm{e}{-}8$), and entropy ablation further reduces performance ($Z = -5.94$, $p < 1.5\mathrm{e}{-}9$). Table 6 shows an additional $1.6\%$ reduction in forgetting rate.

**KMNIST**  On KMNIST (Table 3), "Our(Y)" achieves a $3.3\%$ reduction in MoE error ($Z = -7.09$, $p < 6.7\mathrm{e}{-}13$), while "Our(Z)" yields $2.3\%$ improvement ($Z = -6.48$, $p < 4.7\mathrm{e}{-}11$). Both methods degrade sharply under LB integration ($Z = -39.0$, $p \approx 0$), and entropy ablation also worsens error ($Z = -4.10$, $p < 2.1\mathrm{e}{-}5$). Appendix Table 7 further reports up to a $1.9\%$ reduction in forgetting rate.

**EMNIST (Balanced)**  Table 4 summarizes results on EMNIST. "Our(Y)" reduces MoE error by $12.3\%$ ($Z = -29.6$, $p \approx 0$), while "Our(Z)" achieves $4.1\%$ reduction ($Z = -9.02$, $p < 9.1\mathrm{e}{-}20$). Integration with LB results in severe degradation ($Z = -23.9$, $p \approx 0$), and entropy ablation also harms performance ($Z = -6.89$, $p < 2.7\mathrm{e}{-}12$). Appendix Table 8 further reports a $0.6\%$ reduction in forgetting rate.

**Overall Summary**  Across all four benchmarks, our method consistently outperforms existing baselines in both accuracy and forgetting. Specifically, "Our(Y)" achieves the largest gains, with

| Methods | $k \in \{1,1,1\}$ for $M \in \{1,4,7\}$ | | | $k \in \{1,2,3\}$ for $M \in \{1,4,7\}$ | | | $k \in \{1,4,7\}$ for $M \in \{1,4,7\}$ | | |
|---|---|---|---|---|---|---|---|---|---|
| Vanilla | $8.73_{\pm 0.43}$ | $8.42_{\pm 0.39}$ | $10.8_{\pm 0.73}$ | $8.73_{\pm 0.43}$ | $9.25_{\pm 0.35}$ | $9.84_{\pm 0.53}$ | $8.73_{\pm 0.43}$ | $8.81_{\pm 0.37}$ | $9.70_{\pm 0.31}$ |
| LB | $8.73_{\pm 0.43}$ | $10.6_{\pm 0.61}$ | $10.6_{\pm 0.82}$ | $8.73_{\pm 0.43}$ | $8.35_{\pm 0.49}$ | $9.44_{\pm 0.66}$ | $8.73_{\pm 0.43}$ | $7.99_{\pm 0.27}$ | $8.01_{\pm 0.48}$ |
| ToMoECL | $8.73_{\pm 0.43}$ | $9.76_{\pm 0.46}$ | $11.4_{\pm 0.59}$ | $8.73_{\pm 0.43}$ | $8.18_{\pm 0.56}$ | $8.95_{\pm 0.74}$ | $8.73_{\pm 0.43}$ | $8.01_{\pm 0.28}$ | $8.19_{\pm 0.49}$ |
| ToMoEMEC | $8.73_{\pm 0.43}$ | $8.73_{\pm 0.35}$ | $10.3_{\pm 0.36}$ | $8.73_{\pm 0.43}$ | $8.60_{\pm 0.39}$ | $9.31_{\pm 0.32}$ | $8.73_{\pm 0.43}$ | $8.30_{\pm 0.36}$ | $8.89_{\pm 0.44}$ |
| Our(Y) + LB | $8.73_{\pm 0.43}$ | $10.1_{\pm 0.78}$ | $11.0_{\pm 0.45}$ | $8.73_{\pm 0.43}$ | $9.04_{\pm 0.51}$ | $8.79_{\pm 0.32}$ | $8.73_{\pm 0.43}$ | $7.43_{\pm 0.36}$ | $\underline{7.18}_{\pm 0.52}$ |
| Our(Z) + LB | $8.73_{\pm 0.43}$ | $10.2_{\pm 0.64}$ | $10.3_{\pm 0.65}$ | $8.73_{\pm 0.43}$ | $8.37_{\pm 0.40}$ | $8.51_{\pm 0.88}$ | $8.73_{\pm 0.43}$ | $7.84_{\pm 0.41}$ | $7.94_{\pm 0.69}$ |
| Our(Y) w/o JE | $8.73_{\pm 0.43}$ | $\underline{7.45}_{\pm 0.54}$ | $\underline{7.78}_{\pm 0.35}$ | $8.73_{\pm 0.43}$ | $7.94_{\pm 0.58}$ | $\underline{8.30}_{\pm 0.36}$ | $8.73_{\pm 0.43}$ | $7.42_{\pm 0.22}$ | $7.25_{\pm 0.42}$ |
| Our(Z) w/o JE | $8.73_{\pm 0.43}$ | $8.47_{\pm 0.27}$ | $8.47_{\pm 0.31}$ | $8.73_{\pm 0.43}$ | $7.75_{\pm 0.34}$ | $8.67_{\pm 0.50}$ | $8.73_{\pm 0.43}$ | $8.60_{\pm 0.25}$ | $8.33_{\pm 0.46}$ |
| Our(Y) | $8.73_{\pm 0.43}$ | $\mathbf{6.85}_{\pm 0.49}$ | $\mathbf{6.94}_{\pm 0.35}$ | $8.73_{\pm 0.43}$ | $\mathbf{7.48}_{\pm 0.33}$ | $\mathbf{7.82}_{\pm 0.26}$ | $8.73_{\pm 0.43}$ | $\mathbf{7.36}_{\pm 0.28}$ | $\mathbf{7.11}_{\pm 0.46}$ |
| Our(Z) | $8.73_{\pm 0.43}$ | $7.95_{\pm 0.32}$ | $8.19_{\pm 0.26}$ | $8.73_{\pm 0.43}$ | $\underline{7.62}_{\pm 0.33}$ | $8.36_{\pm 0.40}$ | $8.73_{\pm 0.43}$ | $\underline{7.36}_{\pm 0.46}$ | $7.38_{\pm 0.27}$ |

Table 5: MoE overall forgetting rate (%) on MNIST, 10 runs.

| Methods | $k \in \{1,1,1\}$ for $M \in \{1,4,7\}$ | | | $k \in \{1,2,3\}$ for $M \in \{1,4,7\}$ | | | $k \in \{1,4,7\}$ for $M \in \{1,4,7\}$ | | |
|---|---|---|---|---|---|---|---|---|---|
| Vanilla | $8.43_{\pm 0.79}$ | $10.1_{\pm 1.6}$ | $10.5_{\pm 0.91}$ | $8.43_{\pm 0.79}$ | $9.75_{\pm 1.4}$ | $9.5_{\pm 0.63}$ | $8.43_{\pm 0.79}$ | $10.5_{\pm 1.2}$ | $10.3_{\pm 0.74}$ |
| LB | $8.43_{\pm 0.79}$ | $11.8_{\pm 1.8}$ | $9.11_{\pm 0.80}$ | $8.43_{\pm 0.79}$ | $10.0_{\pm 1.6}$ | $9.38_{\pm 0.66}$ | $8.43_{\pm 0.79}$ | $\underline{9.44}_{\pm 1.4}$ | $8.28_{\pm 0.95}$ |
| ToMoECL | $8.43_{\pm 0.79}$ | $11.9_{\pm 1.7}$ | $9.44_{\pm 0.70}$ | $8.43_{\pm 0.79}$ | $10.5_{\pm 1.6}$ | $9.84_{\pm 0.62}$ | $8.43_{\pm 0.79}$ | $\underline{9.48}_{\pm 1.4}$ | $\underline{8.23}_{\pm 0.94}$ |
| ToMoEMEC | $8.43_{\pm 0.79}$ | $10.5_{\pm 1.6}$ | $10.6_{\pm 0.97}$ | $8.43_{\pm 0.79}$ | $9.76_{\pm 0.66}$ | $10.8_{\pm 0.81}$ | $8.43_{\pm 0.79}$ | $9.54_{\pm 0.60}$ | $11.2_{\pm 0.85}$ |
| Our(Y) + LB | $8.43_{\pm 0.79}$ | $11.9_{\pm 1.7}$ | $9.64_{\pm 0.77}$ | $8.43_{\pm 0.79}$ | $9.69_{\pm 1.4}$ | $9.72_{\pm 0.84}$ | $8.43_{\pm 0.79}$ | $9.94_{\pm 1.4}$ | $8.28_{\pm 0.91}$ |
| Our(Z) + LB | $8.43_{\pm 0.79}$ | $11.7_{\pm 1.7}$ | $9.29_{\pm 1.1}$ | $8.43_{\pm 0.79}$ | $10.2_{\pm 1.5}$ | $9.67_{\pm 0.69}$ | $8.43_{\pm 0.79}$ | $9.94_{\pm 1.1}$ | $9.22_{\pm 1}$ |
| Our(Y) w/o JE | $8.43_{\pm 0.79}$ | $9.39_{\pm 1.4}$ | $\underline{8.54}_{\pm 0.72}$ | $8.43_{\pm 0.79}$ | $\underline{8.43}_{\pm 0.71}$ | $9.22_{\pm 0.91}$ | $8.43_{\pm 0.79}$ | $9.68_{\pm 0.99}$ | $9.73_{\pm 0.79}$ |
| Our(Z) w/o JE | $8.43_{\pm 0.79}$ | $10.8_{\pm 1.1}$ | $\underline{10.6}_{\pm 1.1}$ | $8.43_{\pm 0.79}$ | $9.01_{\pm 0.95}$ | $10.2_{\pm 0.66}$ | $8.43_{\pm 0.79}$ | $12.2_{\pm 1.1}$ | $9.97_{\pm 0.5}$ |
| Our(Y) | $8.43_{\pm 0.79}$ | $9.32_{\pm 1.1}$ | $\mathbf{7.84}_{\pm 0.68}$ | $8.43_{\pm 0.79}$ | $\mathbf{8.00}_{\pm 0.45}$ | $9.14_{\pm 0.82}$ | $8.43_{\pm 0.79}$ | $\mathbf{9.29}_{\pm 1.1}$ | $\mathbf{7.88}_{\pm 0.73}$ |
| Our(Z) | $8.43_{\pm 0.79}$ | $\mathbf{8.49}_{\pm 0.84}$ | $9.16_{\pm 0.57}$ | $8.43_{\pm 0.79}$ | $8.96_{\pm 0.96}$ | $\mathbf{8.97}_{\pm 0.57}$ | $8.43_{\pm 0.79}$ | $10.7_{\pm 1.0}$ | $8.58_{\pm 0.57}$ |

Table 6: MoE overall forgetting rate (%) on Fashion-MNIST, 10 runs.

MoE error reductions of up to $12.3\%$ on EMNIST ($Z = -29.6$, $p \approx 0$) and $9.5\%$ on MNIST ($Z = -15.02$, $p < 2.7\mathrm{e}{-51}$), while "Our(Z)" also provides substantial improvements (up to $8.9\%$, $Z = -12.77$, $p < 1.3\mathrm{e}{-37}$). In contrast, integrating with LB consistently leads to significant degradation (*e.g.*, $Z = -39.0$, $p \approx 0$ on KMNIST), and ablating the joint entropy term results in measurable but smaller performance drops. Forgetting rates are further reduced across all datasets (up to $3.9\%$), confirming the effectiveness of adaptive gating and entropy optimization in maintaining both stability and plasticity. Overall, the results establish the robustness and generality of our approach across diverse continual learning scenarios.

## 5 CONCLUSION

This work presents an information-theoretic framework for optimizing gating in Mixture-of-Experts (MoE) models under the challenging Online Continual Learning (OCL) setting, where each sample is observed only once. Grounded in theoretical analysis, we established a principled link between the achievable Minimum Excess Risk (MER) and the mutual information between expert assignments and labels or outputs. Building on this connection, we proposed two mutual information–based loss functions that support both supervised and unsupervised continual gating, alongside a lightweight matrix-based estimator and a refined joint entropy formulation to ensure efficiency. Comprehensive experiments on standard benchmarks confirm the effectiveness of our approach, achieving up to $12.3\%$ gains in accuracy and $3.9\%$ reductions in forgetting, with strong statistical significance. Overall, our findings underscore the importance of information-driven gating in continual MoE learning and lay a theoretical foundation for advancing dynamic expert routing in more general continual learning scenarios.

## 6 REPRODUCIBILITY STATEMENT

Our source code is submitted as supplementary materials.

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

## A  PROOF OF PROPOSITION 1

*Proof.* For MoE models, the upper bound for MER (Xu & Raginsky, 2022) can be written as

$$\text{MER}_{\ell^{01},t} \leq \sqrt{\tfrac{1}{2} I(\mathcal{Y}_{t,b}; \theta_t \mid \mathcal{X}_{t,b}, \mathcal{G}_{t,b}, \mathcal{D}_{1:t})}. \qquad (9)$$

Focusing on the expert models, we have the Markov data processing chain $\mathcal{G}_{t,b} \leftarrow \mathcal{X}_{t,b} \rightarrow \mathcal{Y}_{t,b}$ and re-arrange it as:

$$\mathcal{G}_{t,b} \rightarrow \mathcal{X}_{t,b} \rightarrow \mathcal{Y}_{t,b}. \qquad (10)$$

Moreover, in the OCL setting, the model parameter $\theta_t$ is trained on $\mathcal{D}_{1:t}$, *i.e.*, $\theta_t = \theta_t | \mathcal{D}_{1:t}$. Given that $\mathcal{G}_{t,b} = g(\mathcal{X}_{t,b})$, we have

$$I(\mathcal{Y}_{t,b}; \theta_t \mid \mathcal{X}_{t,b}, \mathcal{G}_{t,b}, \mathcal{D}_{1:t}) = I(\mathcal{Y}_{t,b}; \theta_t \mid \mathcal{G}_{t,b}).$$

Applying the chain rule of mutual information yields

$$I(\mathcal{Y}_{t,b}; \theta_t \mid \mathcal{G}_{t,b}) = I(\theta_t; \mathcal{Y}_{t,b}) - I(\mathcal{G}_{t,b}; \mathcal{Y}_{t,b}) + I(\mathcal{G}_{t,b}; \mathcal{Y}_{t,b} \mid \theta_t).$$

By the data processing inequality, as shown in Eq. (10), we have

$$I(\mathcal{G}_{t,b}; \mathcal{Y}_{t,b} \mid \theta_t) \leq I(\mathcal{X}_{t,b}; \mathcal{Y}_{t,b} \mid \theta_t).$$

Furthermore, because $(\mathcal{X}_{t,b}, \mathcal{Y}_{t,b})$ is independent of $\theta_t$, it follows that

$$I(\mathcal{X}_{t,b}; \mathcal{Y}_{t,b} \mid \theta_t) = I(\mathcal{X}_{t,b}; \mathcal{Y}_{t,b}).$$

Combining the above, we obtain

$$I(\mathcal{Y}_{t,b}; \theta_t \mid \mathcal{X}_{t,b}, \mathcal{G}_{t,b}, \mathcal{D}_{1:t}) \leq I(\theta_t; \mathcal{Y}_{t,b}) + I(\mathcal{X}_{t,b}; \mathcal{Y}_{t,b}) - I(\mathcal{G}_{t,b}; \mathcal{Y}_{t,b}). \qquad (11)$$

Substituting Eq. (11) into Eq. (9), we conclude

$$\text{MER}_{\ell^{01},t} \leq \sqrt{\tfrac{1}{2} \Big( I(\theta_t; \mathcal{Y}_{t,b}) + I(\mathcal{X}_{t,b}; \mathcal{Y}_{t,b}) - I(\mathcal{G}_{t,b}; \mathcal{Y}_{t,b}) \Big)}.$$

$\square$

## B  RESULTS OF MoE OVERALL FORGETTING RATE

| Methods | $k \in \{1,1,1\}$ for $M \in \{1,4,7\}$ | | | $k \in \{1,2,3\}$ for $M \in \{1,4,7\}$ | | | $k \in \{1,4,7\}$ for $M \in \{1,4,7\}$ | | |
|---|---|---|---|---|---|---|---|---|---|
| Vanilla | $13.2_{\pm0.6}$ | $13.4_{\pm0.67}$ | $15.6_{\pm0.76}$ | $13.2_{\pm0.6}$ | $13.3_{\pm0.60}$ | $14.3_{\pm0.64}$ | $13.2_{\pm0.6}$ | $13.6_{\pm0.43}$ | $13.9_{\pm0.73}$ |
| LB | $13.2_{\pm0.6}$ | $11.4_{\pm0.34}$ | $11.7_{\pm0.52}$ | $13.2_{\pm0.6}$ | $9.92_{\pm0.43}$ | $11.2_{\pm0.66}$ | $13.2_{\pm0.6}$ | $9.69_{\pm0.49}$ | $10.4_{\pm0.83}$ |
| ToMoECL | $13.2_{\pm0.6}$ | $11.6_{\pm0.32}$ | $11.6_{\pm0.59}$ | $13.2_{\pm0.6}$ | $10.3_{\pm0.51}$ | $11.1_{\pm0.64}$ | $13.2_{\pm0.6}$ | $9.58_{\pm0.46}$ | $10.6_{\pm0.82}$ |
| ToMoEMEC | $13.2_{\pm0.6}$ | $14.4_{\pm0.35}$ | $14.9_{\pm0.74}$ | $13.2_{\pm0.6}$ | $12.7_{\pm0.44}$ | $14.1_{\pm0.45}$ | $13.2_{\pm0.6}$ | $13.5_{\pm0.50}$ | $13.7_{\pm0.58}$ |
| Our(Y) + LB | $13.2_{\pm0.6}$ | $11.3_{\pm0.35}$ | $11.8_{\pm0.57}$ | $13.2_{\pm0.6}$ | $9.94_{\pm0.46}$ | $11.1_{\pm0.61}$ | $13.2_{\pm0.6}$ | $9.99_{\pm0.45}$ | $10.1_{\pm0.70}$ |
| Our(Z) + LB | $13.2_{\pm0.6}$ | $11.6_{\pm0.41}$ | $11.7_{\pm0.44}$ | $13.2_{\pm0.6}$ | $10.2_{\pm0.44}$ | $11.1_{\pm0.67}$ | $13.2_{\pm0.6}$ | $10.2_{\pm0.53}$ | $10.3_{\pm0.84}$ |
| Our(Y) w/o JE | $13.2_{\pm0.6}$ | $14.8_{\pm0.29}$ | $15.5_{\pm0.74}$ | $13.2_{\pm0.6}$ | $14.0_{\pm0.35}$ | $14.1_{\pm0.62}$ | $13.2_{\pm0.6}$ | $14.4_{\pm0.56}$ | $13.5_{\pm0.66}$ |
| Our(Z) w/o JE | $13.2_{\pm0.6}$ | $14.9_{\pm0.58}$ | $15.2_{\pm0.42}$ | $13.2_{\pm0.6}$ | $13.5_{\pm0.64}$ | $13.9_{\pm0.43}$ | $13.2_{\pm0.6}$ | $12.4_{\pm0.65}$ | $13.4_{\pm0.59}$ |
| Our(Y) | $13.2_{\pm0.6}$ | $14.2_{\pm0.53}$ | $14.4_{\pm0.80}$ | $13.2_{\pm0.6}$ | $13.8_{\pm0.27}$ | $13.9_{\pm0.33}$ | $13.2_{\pm0.6}$ | $14.1_{\pm0.58}$ | $13.2_{\pm0.78}$ |
| Our(Z) | $13.2_{\pm0.6}$ | $14.2_{\pm0.35}$ | $14.8_{\pm0.51}$ | $13.2_{\pm0.6}$ | $13.4_{\pm0.33}$ | $13.5_{\pm0.52}$ | $13.2_{\pm0.6}$ | $11.7_{\pm0.41}$ | $13.2_{\pm0.54}$ |

Table 7: MoE overall forgetting rate (%) on KMNIST, 10 runs.

| Methods | $k \in \{1,1,1\}$ for $M \in \{1,4,7\}$ | | | $k \in \{1,2,3\}$ for $M \in \{1,4,7\}$ | | | $k \in \{1,4,7\}$ for $M \in \{1,4,7\}$ | | |
|---|---|---|---|---|---|---|---|---|---|
| Vanilla | $7.98_{\pm0.26}$ | $10.0_{\pm0.44}$ | $9.68_{\pm0.40}$ | $7.98_{\pm0.26}$ | $10.2_{\pm0.30}$ | $10.1_{\pm0.38}$ | $7.98_{\pm0.26}$ | $9.68_{\pm0.24}$ | $10.33_{\pm0.37}$ |
| LB | $7.98_{\pm0.26}$ | $8.32_{\pm0.46}$ | $8.50_{\pm0.20}$ | $7.98_{\pm0.26}$ | $7.11_{\pm0.25}$ | $6.63_{\pm0.29}$ | $7.98_{\pm0.26}$ | $7.01_{\pm0.12}$ | $7.22_{\pm0.10}$ |
| ToMoECL | $7.98_{\pm0.26}$ | $8.69_{\pm0.37}$ | $8.72_{\pm0.20}$ | $7.98_{\pm0.26}$ | $6.72_{\pm0.32}$ | $7.25_{\pm0.57}$ | $7.98_{\pm0.26}$ | $7.01_{\pm0.12}$ | $7.23_{\pm0.10}$ |
| ToMoEMEC | $7.98_{\pm0.26}$ | $10.2_{\pm0.42}$ | $11.1_{\pm0.54}$ | $7.98_{\pm0.26}$ | $9.40_{\pm0.25}$ | $11.3_{\pm0.33}$ | $7.98_{\pm0.26}$ | $9.63_{\pm0.24}$ | $10.43_{\pm0.34}$ |
| Our(Y) + LB | $7.98_{\pm0.26}$ | $8.55_{\pm0.36}$ | $8.43_{\pm0.18}$ | $7.98_{\pm0.26}$ | $6.53_{\pm0.27}$ | $7.27_{\pm0.47}$ | $7.98_{\pm0.26}$ | $7.08_{\pm0.14}$ | $7.13_{\pm0.13}$ |
| Our(Z) + LB | $7.98_{\pm0.26}$ | $8.54_{\pm0.38}$ | $8.63_{\pm0.23}$ | $7.98_{\pm0.26}$ | $7.01_{\pm0.41}$ | $6.75_{\pm0.53}$ | $7.98_{\pm0.26}$ | $7.14_{\pm0.15}$ | $7.16_{\pm0.15}$ |
| Our(Y) w/o JE | $7.98_{\pm0.26}$ | $12.8_{\pm0.34}$ | $15.4_{\pm0.45}$ | $7.98_{\pm0.26}$ | $10.9_{\pm0.34}$ | $12.2_{\pm0.37}$ | $7.98_{\pm0.26}$ | $9.49_{\pm0.44}$ | $9.97_{\pm0.30}$ |
| Our(Z) w/o JE | $7.98_{\pm0.26}$ | $11.1_{\pm0.44}$ | $11.0_{\pm0.62}$ | $7.98_{\pm0.26}$ | $10.3_{\pm0.62}$ | $11.5_{\pm0.55}$ | $7.98_{\pm0.26}$ | $9.35_{\pm0.49}$ | $9.82_{\pm0.33}$ |
| Our(Y) | $7.98_{\pm0.26}$ | $12.5_{\pm0.44}$ | $14.7_{\pm0.62}$ | $7.98_{\pm0.26}$ | $10.7_{\pm0.37}$ | $11.1_{\pm0.41}$ | $7.98_{\pm0.26}$ | $9.24_{\pm0.50}$ | $9.96_{\pm0.16}$ |
| Our(Z) | $7.98_{\pm0.26}$ | $10.3_{\pm0.48}$ | $11.0_{\pm0.20}$ | $7.98_{\pm0.26}$ | $10.3_{\pm0.53}$ | $10.8_{\pm0.59}$ | $7.98_{\pm0.26}$ | $9.12_{\pm0.34}$ | $9.78_{\pm0.39}$ |

Table 8: MoE overall forgetting rate (%) on EMNIST (Balanced), 10 runs.

## C  THE USE OF LARGE LANGUAGE MODELS

We use Large Language Models to polish writing as a general-purpose assist tool.

