# OpenReview forum: "Learning Once, Routing Right: Information-Theoretic Gating for Online Continual Mixture-of-Experts"
_ICLR.cc/2026/Conference — ICLR 2026 Conference Withdrawn Submission_

### Official Review · Reviewer_s9jm · 2025-10-19

**Soundness:** 2
**Presentation:** 3
**Contribution:** 2
**Rating:** 2
**Confidence:** 4

**Summary:**

The paper introduces an information-theoretic framework for optimizing gating mechanisms in Mixture-of-Experts (MoE) models under online continual learning. The central theoretical claim is that the Minimum Excess Risk (MER) of MoE models can be upper-bounded by terms involving mutual information between expert assignments and targets (labels or outputs). This connection motivates the design of two mutual information–based loss functions, applicable to both supervised and unsupervised continual learning. To make mutual information computation tractable, the authors propose a lightweight, matrix-based estimator with an improved joint entropy formulation. Experiments on MNIST, Fashion-MNIST, KMNIST, and EMNIST show consistent performance improvements.

**Strengths:**

1. This paper is well-written and easy to follow.

2. The proposed theoretical analysis of mutual information between expert assignments and labels/outputs is interesting to understand MoE in continual learning.

**Weaknesses:**

1. The proposed method is only evaluated on vary simple datasets, such as MNIST as its variants.

2. The comparison baselines are also very simple, without considering sufficiently representative methods in continual learning.

3. The proposed method seems to achieve very marginal improvements over the simple baselines.

4. Do the theoretical analysis and the proposed method only apply to online continual learning? Is it possible to extend them to broader continual learning settings (e.g., offline continual learning)?

**Questions:**

My major concerns lie in the comparison baselines and applicability of the proposed method. Please refer to the Weaknesses.

---

### Official Review · Reviewer_pQyN · 2025-10-26

**Soundness:** 3
**Presentation:** 3
**Contribution:** 3
**Rating:** 6
**Confidence:** 4

**Summary:**

This paper proposes a new theoretical framework for optimizing the gating mechanism in Mixture-of-Experts (MoE) models for the Online Continual Learning (OCL) setting. The core contribution is a novel theoretical link between the Minimum Excess Risk (MER) of the MoE model and the mutual information (MI) between the expert assignments and the labels/outputs. Building on this theory, the authors derive two MI-based loss functions to optimize the gating network, one for labeled data and one for a label-free scenario. To make this practical, they also propose a lightweight, matrix-based MI estimator. The method is evaluated on a suite of MNIST-variant datasets (MNIST, Fashion-MNIST, KMNIST, EMNIST), where it shows improvements over other MoE-based baselines.

**Strengths:**

1. The paper is well-written and clearly motivated. The authors do a good job of identifying a specific gap—the lack of a principled, theoretical foundation for MoE gating in continual learning—and proposing a solution.

2. The primary strength is the theoretical connection between MER and mutual information. Framing the gating optimization as an MI maximization problem derived from a more fundamental concept (MER) is a principled and elegant approach, moving beyond common heuristics like load-balancing.

3. The authors thoughtfully bridge the gap from their theory to a practical algorithm by developing a lightweight matrix-based MI estimator. This shows a good understanding of the practical limitations of information-theoretic objectives and makes the work self-contained.

**Weaknesses:**

1. My main concern is the limited scope of the experiments. The paper relies exclusively on MNIST-variant datasets (MNIST, F-MNIST, KMNIST, EMNIST). While these are acceptable for a preliminary "proof-of-concept" of a theoretical idea, they are relatively simple and not representative of the complex challenges in modern, large-scale CL. The community has largely moved to more challenging benchmarks (e.g., CIFAR-100, TinyImageNet, DomainNet) to demonstrate state-of-the-art performance.

2. Related to the first point, the set of baselines (ToMoECL, ToMoEMEC, LB) appears to be limited to other MoE-gating or theoretical papers. It is difficult to situate the performance of this method within the broader CL landscape. The paper would be significantly stronger if it included comparisons against standard, widely-recognized CL methods (e.g., key replay-based, regularization-based, or architectural methods), even if they are not MoE-based.

3. The choice of MER as the starting point is novel, but it isn't a standard framework for analyzing continual learning. The paper would benefit from a brief discussion justifying why MER is the most appropriate theoretical lens for this problem, as opposed to other common theoretical frameworks used in CL or online learning.

**Questions:**

1. Could the authors comment on the scalability of their method? How do they expect the MI-based loss and its estimator to perform on more complex, higher-dimensional problems like CIFAR-100 or ImageNet-R, where the underlying distributions are far more complex than MNIST?

2. Why were more standard CL baselines (e.g., EWC, GEM, ER) omitted? Even if the method is MoE-specific, these baselines would provide a crucial performance anchor to understand the practical benefit of the proposed approach.

---

### Official Review · Reviewer_yedf · 2025-11-07

**Soundness:** 2
**Presentation:** 1
**Contribution:** 1
**Rating:** 2
**Confidence:** 3

**Summary:**

This paper investigates the relationship between Minimum Excess Risk (MER) and mutual information, and proposes two new gating functions that minimize MER through mutual information maximization. It also introduces a lightweight method to reduce the computational complexity of mutual information estimation. The experimental results demonstrate the superior performance of the proposed design.

**Strengths:**

1. They studied MoE of CL from the perspective of Minimum Excess Risk, which is interesting.

2. They provide a series of experiments to explain the advantage of their proposed loss design.

**Weaknesses:**

1. The presentation is not clear to me. The notations they use are not defined well. Detailed questions are provided in "Questions".

2. Though they state that "we develop a rigorous theoretical framework to analyze MoE models in the OCL setting", there are a lot of statement in this paper is heoristic, e.g., from L240 - L252. Detail question are provided in "Questions".

**Questions:**

Some definition of the core idea in this paper is not clear to me:
1. In proposition 1, which is your main result, what is $\psi$? Does it have an explicit expression?
2. In L219 - L222, what is the reason of designing such a form? Why you need to normalize by $H$ rather than by dividing (norm of) $I$ itself? How to choose $\gamma_y$ and $\gamma_z$?

Oher questions:
1. To explain the relationship between MER and mutual information, the MER is defined in L187 - L191. My question is why this MER is related to the performance of continual MoE?
2. What is the purpose of proposing $H'$ in eq(7)? In my understanding, it is to solve the issue "H_\alpha(A, A) ≥H_\alpha(A)". But why it is an issue? If you train the model by using eq(6), will it influence the training result heavily?
3. Can you explain that why your proposed method in 3.3 reduces time complexity?

Other question about presentation and definition of this paper:
1. In L144, what is the definition of ⊙?
2. In L197, what is the definition of I(.,.)?
3. In L210 - L212, how did you define "Markov data processing chain"?
4. In L102 - L103, the citation Li et al. (2025) and  Li & Duan (2025) refers to the same paper.

---

### Note · Authors · 2025-11-19

I have read and agree with the venue's withdrawal policy on behalf of myself and my co-authors.